# *Corythauma ayyari* (Insecta, Heteroptera, Tingidae) depends on its host plant to spread in Europe

**Manon Durand, Eric Guilbert** *

UMR7179 CNRS-MNHN, National Museum of Natural History, Paris, France

* eric.guilbert@mnhn.fr

## Abstract

Biological invasions increase with the intensity of globalization, human activities, and climate change. Insects represent a high potential of invasive species due to their adaptability to new environment. We analysed here the potential of an Asian phytophagous bug, *Corythauma ayyari* (Heteroptera, Tingidae) to become widespread, recently recorded in Europe, and that depends on *Jasminum* spp., an ornamental plant widespread in Europe. We modelled its current distribution, projected it into the future and tested its niche overlap between native and invaded areas. When considering the host plants as environmental variables, the analysis shows that *C. ayyari* shifted to a new ecological niche but its distribution is restricted by its host plant distribution. Including or excluding the host plants as environmental variables has an impact on *C. ayyari* distribution. We recommend to consider host plant interactions when dealing with niche modelling of phytophagous species.

## Introduction

An invasive species is an non-native species that begins to proliferate in a new geographical habitat. As such, it usually becomes a pest. A pest is a native or non-native species disrupting the environment. The invasion is often characterized by an introduction, a naturalization and a spread [1] and its success is context dependent [2].

While the number of biological invasions continues to increase, particularly with the intensity of globalization and the acceleration of climate change, they are now one of the main causes of the current loss of biodiversity. Trade would be the major driver of alien species introductions at least for invertebrates and algae [3]. In order to fight against their proliferation and the damages, it is necessary to identify these species and to analyse their invasiveness.

Insects are the most diverse group of living species on earth, thanks in part to their ability to adapt to new environmental conditions. It is therefore not surprising that many insect species are considered invasive or potentially invasive. Insects are in fact the most disrupting class of invasive species, spreading diseases, consuming crops, disrupting forest ecosystem [4, 5]. For example, 50% of recently invasive species belong to the order Hemiptera [6]. *Corythauma ayyari* (Drake) belongs to the family Tingidae (Hemiptera, Heteroptera). It has been first

**Data Availability Statement:** All relevant data are within the manuscript and its Supporting Information files.

**Funding:** The author(s) received no specific funding for this work.

**Competing interests:** The authors have declared that no competing interests exist.

described from India (Drake 1933) and found on *Jasminum pubescens* (Retz.), today synonymized with *J. multiflorum* (Burm.). It is notably found in Pakistan, Sri Lanka, Laos, Thailand, Malaysia, India, Indonesia, and Singapore [6–8]. It is known as a pest of Jasmine in Southern India [9, 10]. However, very few specimens were recorded from these areas, as most collecting were casual. Recent observations report its presence in France [8, 11], Italy [7, 8, 12], Spain [6], Tunisia [13], Israel [14], and Syria [15]. Most records mention the three main host plants *Jasminum officinale* L., *J. grandiflorum* L., and *J. sambac* L. (Oleaceae) but also *Volkameria inermis* L. (Lamiaceae) [15] and *Trachelospermum* sp. (Apocynaceae) [11].

It is hypothesized to have been accidentally introduced in Europe several times independently [8] probably with *Jasminum* plants widely used in public and private gardens, as they are highly valued for their fragrance [6, 8]. However, the maintenance of *C. ayyari* on Jasmine species is not without damage to the host plant. It has been observed that adults and nymphs develop on the underside of the leaves and feed on sap. The excreta of the tingid mar the leaves, greatly decreasing photosynthesis, which deteriorates the palisade parenchyma and leads to progressive desiccation of the plant [6–8, 12, 15]. The recent arrival in Europe of *C. ayyari* suggests that it could wreak havoc on Jasmine species but also other ornamental and/or crop plants such as *Volkameria* or *Trachelospermum* species, causing significant environmental and economic damage.

Species Distribution Models (SDM) are increasingly used to answer biogeographic and ecological questions. Indeed, these models allow to estimate the probability of presence of a species according to different biotic and abiotic variables, in order to expose the favorable areas for its presence [16, 17]. These tools are particularly interesting when studying invasive or potentially invasive species. Previous studies have indeed demonstrated the interest and reliability of these methods to quantify the probability of presence and niche differences of an invasive species [18, 19]. Niche differences can be easily detected when variables driving species distribution are known [18], which is particularly relevant when dealing with invasive species. Close match was found between habitat suitability in potential invaded areas and pattern of occurrences of invasive species using bioclimatic niche modelling [19].

Phytophagous insects such as Tingidae which are sap suckers, are strictly dependent of their host plant. Particularly if they are monophagous. Therefore, their distribution highly depends on the distribution of the host plant. Two publications have already focused on the current distribution of *C. ayyari* at the level of the Mediterranean basin and specially in the Iberian Peninsula [6] and in Italy [8]. Therefore, it makes sense to reinforce them by focusing here on the invasive potential of the species.

The aim of this study is to analyse the invasive potential of *C. ayyari* by modelling its current distribution and its potential ecological niche in Europe, according to the current climatic conditions and the distribution of its main host plants. We modelize *C. ayyari* distribution including or not the four main host plants (*J. officinale*, *J. multiflorum*, *J. grandiflorum* and *J. sambac*), as environmental variable to evaluate their influence on *C. ayyari* distribution. On this basis, *C. ayyari* potential distribution is projected into the future. The native and the invasive ecological niches are compared to see if a shift from native to invasive niche is observed.

## Materials & methods

### Study site

The study focuses on the potential spread area of *C. ayyari*, according to the distribution of its host plants and considering their native area in Asia. It includes an area between 18˚ West to 141˚ East and between 1˚ to 70˚ North; covering the distribution area of *Jasminum* species in Europe and the Mediterranean Basin between 18˚ West to 40˚ East and 30˚ to 70˚ North, and

the native area which covers a surface within 1˚ to 46˚ North and 65˚ to 141˚ East. It covers also part of Africa, where occurrences of *Jasminum* species are known. If occurrences of the host plants are known from North America, none of *C. ayyari* is known there.

## Species occurrences

Occurrences of *C. ayyari* were retrieved from the GBIF network (dio:10.15468/dl.b3jj26), from scientific publications [8, 11, 13, 20, 21] and also from personal communications. Many other specimens of *C. ayyari* are known but were collected long time ago and without accurate location as to be used in this study. Similarly, occurrences of *J. officinale*, *J. grandiflorum*, *J. multiflorum*, and *J. sambac*, were extracted from the GBIF network (dio:10.15468/dl.9sbc2q and dio:10.15468/dl.2j7zqa). *Jasminum officinale* has an extended distribution in Europe, compared to the three other *Jasminum* species. There are only 21 records of *J. grandiflorum* in South West Europe, the rest are located in Asia. *Jasminum multiflorum* is sparcely recorded from South East Asia and totalizes two records in Europe (Germany and Greece). *Jasminum sambac* is also widely distributed in Asia but is mainly recorded in India. There are only five records of *J. sambac* in North-East Europe. We did not include *Trachelospermum* sp. as host plant, as the species has not been identified, and *Volkameria inermis* as it is a tropical species, essentially distributed on the seashore in South East Asia between India and the South Pacific islands. No field survey was done in this study to acquire original data. As such no permits were necessary.

## Environmental variables

We used the 19 climatic variables available in WorldClim version 2.0 (www.worldclim.org) at a spatial resolution of 2.5 minutes. We also used the potential distribution of the four *Jasminum* species as biotic variable for *C. ayyari*. The procedure is explained below.

## Niche modelling

Before modelling the distribution of species, we analyse the correlation between variable using 'Hmisc' package and Pearson coefficient, either for the current and the future conditions. The sets of current and future climatic variables with a correlation coefficient up to 0.7 were different. As collinearity affects models trained on data at different times, we opted for variables issued from a principal component analysis (PCA) [22]. We did a PCA of the 19 climatic variables either of the current and future conditions using 'FactoMineR' package, and we selected the first nine PCA components as variables, accounting for 99% of the variance. *Jasminum* species modelled distributions were also used as biotic variables as they were not correlated up to 0.7 to the PCA components used as variables, and as *Jasminum* species and *C. ayyari* distributions were not correlated.

 Data were cleaned before modelling using 'CoordinateCleaner' package [23], to get rid of inadequate occurrences such as the specimens deposited in museums, occurrences wrongly geo-referenced or duplicated. We considered the tests 'urban_ref', 'capitals_ref', as null since *Jasminum* species are ornamental plants and therefore can be present in urban environments.

 Pseudo-absences were added to the dataset of occurrences. We selected the same amount of pseudo-absences as occurrences as suggested by Bardet-Massin et al. [16]. We randomly selected 10 sets of pseudo-absences for each *Jasminum* species, assuming that their distribution is in equilibrium since they were introduced in Europe and the Mediterranean Basin long time ago. Assuming that the distribution of *C. ayyari* is not in equilibrium [24] as we deal with an invasive species, we selected pseudo-absence out of a convex hull calculated on the basis of the known occurrences [25]. According to several authors, low sample size does not affect model

accuracy as soon as the species have small geographic range and restricted environmental tolerance [26, 27], which seems to be the case here. In addition, each dataset analysed here was up to 20–30 observations, which is the minimum sample size required to ensure model accuracy under generic procedures of modelling as stated by Jiménez-Valverde [28].

In a first step, we modelized the potential distribution of *Jasminum* species considering PCA variables, and projected it into the future (2081–2100 period). In a second step, we did a first modelling of *C. ayyari* potential distribution, considering PCA variables only, and we did a second one considering PCA variables and including the rasters resulting of the modelled distributions of *Jasminum* species as biotic variables. Prior to this last analysis, we tested the correlation between *C. ayyari* distribution and the one of *Jasminum* species. The two modelling were compared to see the influence of *Jasminum* species distribution on *C. ayyari* modelling. We projected the distribution of *C. ayyari* into the 2081–2100 period as well, using only PCA variables corresponding to the period, and in a second step including also the modelled projection into the future of *Jasminum* species. For the projection into the future, we considered SSP585 scenario, the worst scenario of global warming, corresponding to an additional radiative forcing of 8.5 W/m$^2$, and we use the CNRM-CM6-1 model, which presents particularly fine parameter calibration strategies tailored to the geographic area considered [29]. Modelling was made using 'Biomod2' package [30] and the following options:

All the models were tried except Maxent which is the only one based on presence only, i.e. Generalized Linear Model (GLM), Generalized Additive Model (GAM), Generalized Boosting Model (GBM), Classification Tree Analysis (CTA), Artificial Neural Network (ANN), Surface Range Envelop (SRE), Flexible Discriminant Analysis (FDA), Random Forest (RF), and Multiple Adaptive Regression Splines (MARS).

We run 10 evaluations, split the data in 20% test data and 80% modelling data. Models were selected to be combined using a threshold of 0.7 on the basis of TSS evaluation. Modelling was evaluated by ensemble forecasting, combining all model as suggested by [31], with an arbitrary threshold up to 0.7 and using the TSS binary metric which is independent to prevalence, as recommended by [32]. Combined model was evaluated using Boyce index with 'ecospat' package [33, 34]. Boyce index measures how much model predictions differ from random distribution of the observed distribution. It ranges between -1 and +1. A value of +1 indicates the model fits perfectly with the distribution and a value of zero indicates the model is not different from a random distribution.

## Niche comparison

We also compared ecological niches of *Jasmium* species and *C. ayyari* in the invaded area. Ecological niche estimates were performed with the 'ecospat' package [34, 35]. A Mantel test was done first to test spatial autocorrelation between variables. Niche overlap between native and invaded areas was also measured using the Schoener's D index [35–37]. It calculates the occupancy of the environment by the species. The comparison of this occupancy between two areas or two species is used to calculate the niche overlap. It can range from 0, for niches with no overlap, to 1, for completely identical niches. A niche equivalency and a niche similarity test are performed. The equivalency tests whether the niche overlap is constant when randomly reallocating the occurrences of two species among two ranges. The similarity tests whether the occupied niche in one range is more similar to the one occupied in the other range than expected by random. We evaluated niche overlap on the basis of the 19 climatic variables. We compare niche overlap only for *C. ayyari*, *J. grandiflorum* and *J. officinale*, as the two other *Jasminum* species do not have enough occurrences in the invaded area to be analysed.

## Results

*Jasminum officinale* totalizes 1381 occurrences restricted to the areas studied, of which 78% are located in Europe. It totalizes 1065 occurrences after cleaning. *Jasminum sambac* totalizes 626 occurrences after cleaning over 724 existing. *Jasminum grandiflorum* totalizes 186 occurrences after cleaning over 203 existing, and *J.multiflorum* totalizes 27 occurrences after cleaning over 40 existing. *Corythauma ayyari* totalizes 48 occurrences, of which 37 are located in Europe and 11 are located in Asia. It totalizes 34 occurrences after cleaning (S1 Data) (Fig 1).

### Niche modelling

The results of current distribution modelling and projection into the future of *Jasminum* species is illustrated in supplementary information (S1 Fig). Boyce index is high for *J. officinale* (0.952), *J. grandiflorum* (0.946) but low for *J. multiflorum* (0.65), and *J. sambac* (0.668). Including *Jasminum* distributions as variable, Boyce index for *C. ayyari* distribution modelling is 0.760; while without *Jasminum* variables it is 0.659.

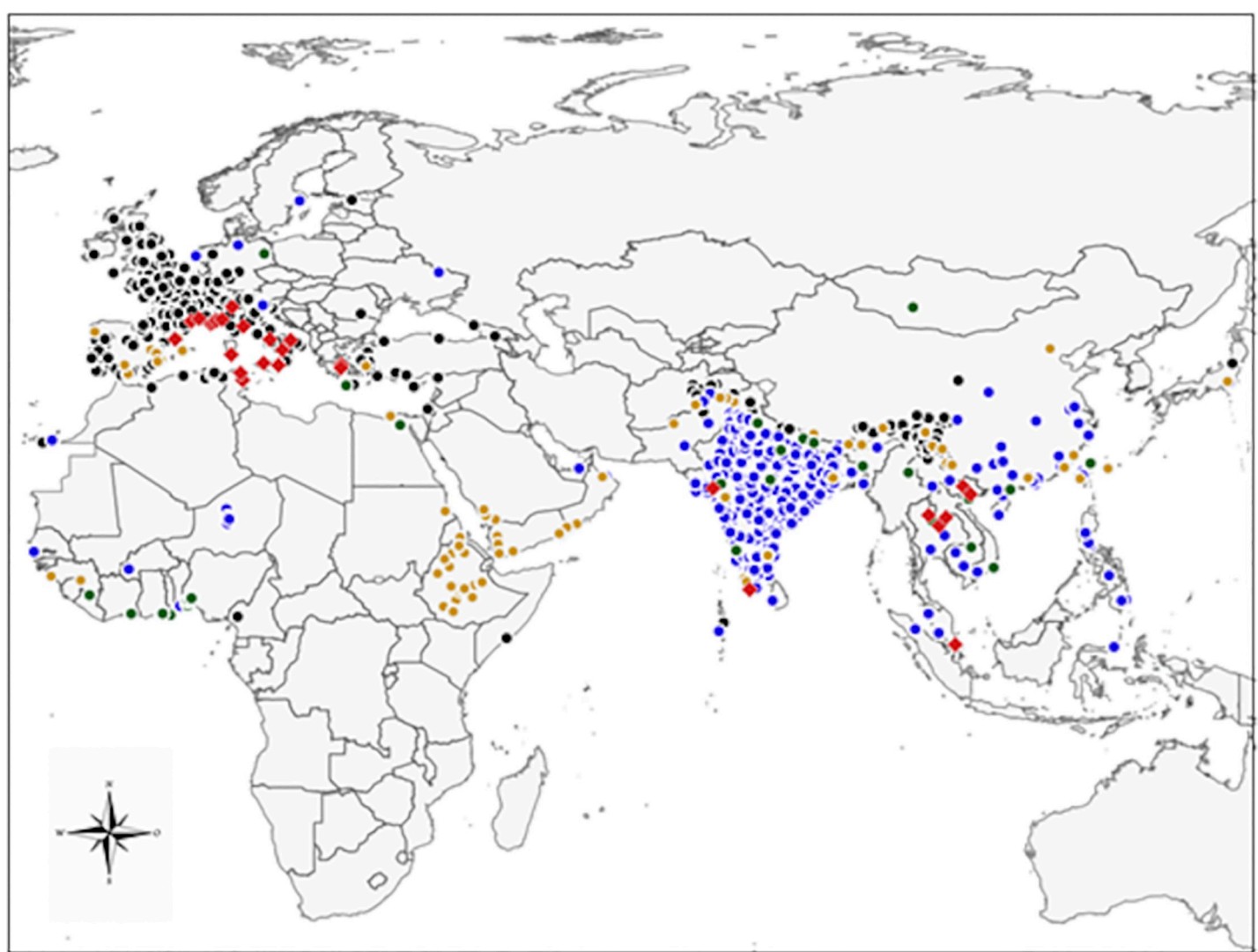

**Fig 1. Map of the occurrences used.** Map of the study area showing the occurrences of *C. ayyari* (red diamonds), and its four main host plants, *J. officinale* (black dots), *J. grandiflorum* (orange dots), *J. multiflorum* (green dots), and *J. sambac* (blue dots).

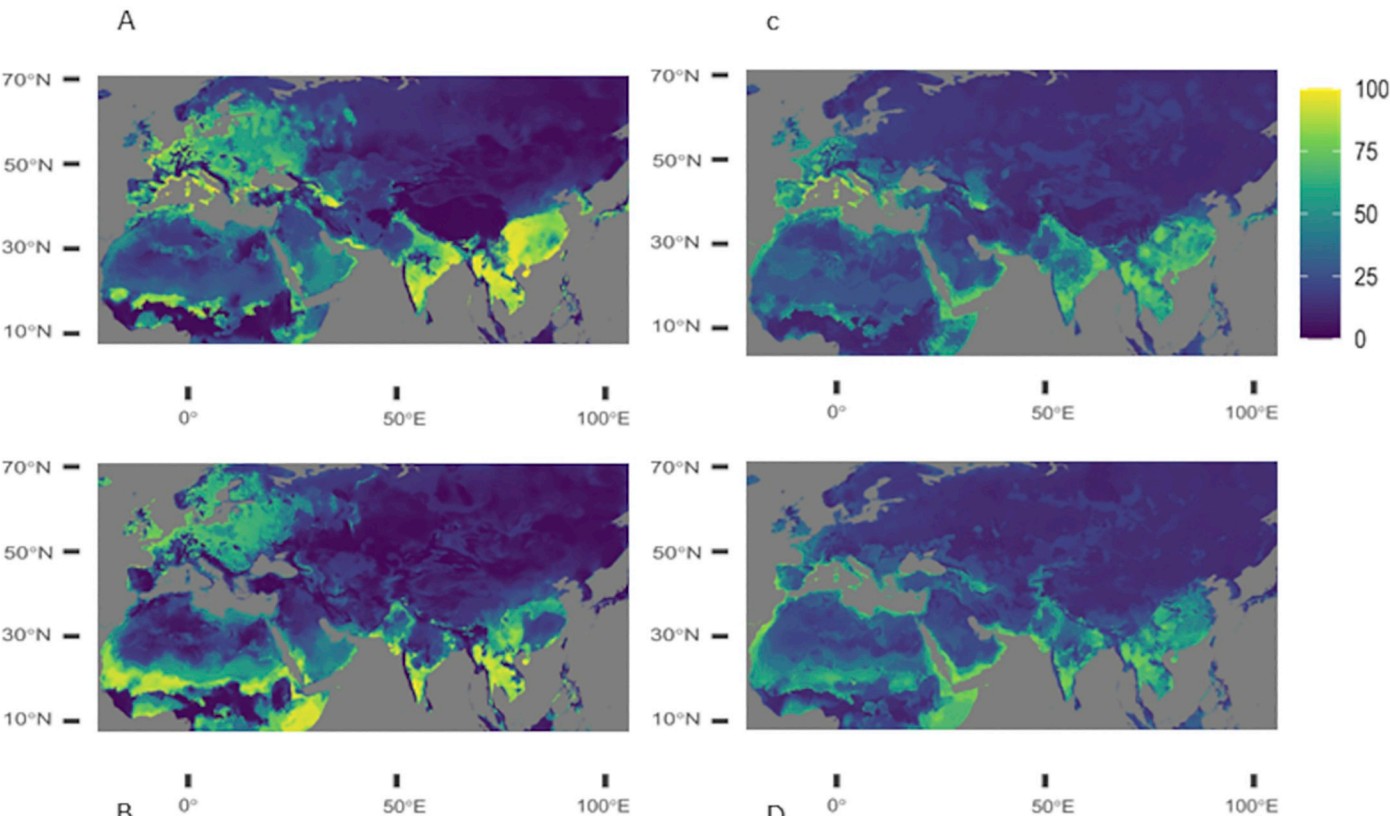

**Fig 2. Projections of the current modeling and future projection of C. ayyari distribution.** Projections of the current (upper row) and future (lower row) environmental conditions for *C. ayyari* using climatic variables only (left colon), and including also *Jasminum* species distribution as variables (right colon). The scale represents percentage of suitability along a gradient of colors.

Considering only PCA components, *Corythauma ayyari* current distribution modelling shows a high suitable habitat on the North Mediterranean coasts in Spain, France, Italy, the Croatian coasts, the Aegean and Marmara coasts, the South of Black Sea coast, part of North of France, Belgium and the Netherlands, at a lower degree parts of North and Central Europe except some central areas, and North Scandinavia. In Asia, suitable areas include India except West coast and North, Thailand, Myanmar, Laos, Vietnam and Cambodgia except West coast of South East Asia, Malaysia included, and a large part of South and East China (Fig 2A).

*Corythauma ayyari* distribution projected into the future shows a reduction of the current suitable areas mainly Spain, Italia, the North of Mediterranean and Marmara coasts, also moving North of Europe facing the Channel and the North Sea, parts of Scandinavia where some unsuitable areas turned 50% suitable (Fig 2B). In Asia, East China and most areas North to India turned unsuitable.

Correlation between *Jasminum* species distribution and *C. ayyari* distribution are low (ranging between 0.32 to 0.46, S1 Table). Correlation between *Jasminum* species does not exceed 0.53 (between *J. multiflorum* and *J. grandiflorum*), lesser than the usual threshold accepted for correlated variables in niche modelling which is 0.7. As such, including *Jasminum* species distribution as variables does not bias modelling.

When including *Jasminum* species distribution as variables, *C. ayyari* current distribution modelling shows suitable areas reduced to North Mediterranean coasts, including Aegean and Marmara coasts. It includes also part of West Europe and Maghreb coasts with a lesser

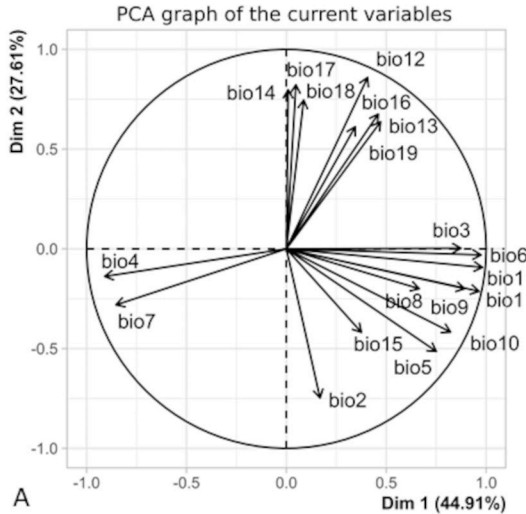
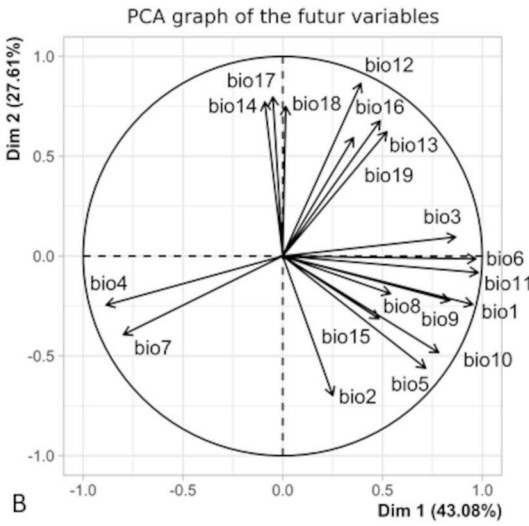

**Fig 3.** PCA plot of the 19 bioclimatic variables on the first two axes, A) for the current conditions, and B) for the future conditions. The graphs show the 19 bioclimatic variables contribution on the two first axes of the PCA for the current conditions and for the future conditions (2081–2100).

suitability. In Asia, areas are not highly suitable, however, they include Part of South and East India, central South East Asia, except Malaysia and a large region in South East China (Fig 2C).

*Corythauma ayyari* distribution projected into the future with Jasminum species distributions reduces suitable areas to a thin range along Mediterranean coasts, enlarging suitable areas to most of South Mediterranean coasts, Portugal, the Atlantic side of Spain and South West of France. Suitability of areas is reduced in Asia, except in West India in Gujarat region and Pakistan (Fig 2D).

The most contributing variables for *C. ayyari* modelling for the ensemble forecasting are the 5th, 1st and 2nd PCA components (39%, 15% and 12%, respectively). When including *Jasminum* distribution as variables, the three most contributing variables are the 5th PCA component (28%) and *J. officinale* and *J. grandiflorum* distribution modeling (23% and 13%, respectively) (S2 Table, Fig 3A and 3B).

## Niche comparison

Overlap of *Jasminum* species and *C. ayyari* niches in the native area is low between all species (Table 1), except between *J. multiflorum* and *J. grandiflorum* (70%), for which the similarity is significant.

**Table 1. Niche overlap, equivalency and similarity tests between *Jasminum* species and C. ayyari in the native area.**

|  | Overlap | Similarity | Equivalency |
|---|---|---|---|
| *J. officinale* | 0.0707 | 0.1287 | 1 |
| *J. grandiflorum* | 0.0156 | 0.3762 | 1 |
| *J. multiflorum* | 0.0329 | 0.3168 | 0.6363 |
| *J. sambac* | 0.1138 | 0.1089 | 0.5454 |

Niche overlap, similarity and equivalency test between *Jasminum* species and *C. ayyari* in the native area.

**Table 2. Niche overlap, equivalency and similarity tests between species in the invaded area.**

|  | Overlap | Similarity | Equivalency |
|---|---|---|---|
| *J. officinale* | 0.2726 | *0.0297* | 1 |
| *J. grandiflorum* | 0.0680 | 0.1881 | 0.9091 |

Niche overlap, similarity and equivalency test between *Jasminum* species and *C. ayyari* in the invaded area. Significant values are in italic.

Overlap between species in the invaded niche is significantly similar only between *J. officinale* and *C. ayyari*. It has not been tested for *J. multiflorum* and *J. sambac* due to the too low number of occurrences in Europe (Table 2).

Comparison of native and invaded niche for *C. ayyari* shows a very low overlap (D = 0.024). It is neither equivalent, nor similar (P value = 0.9604 and 0.0594, respectively).

## Discussion

The biology of *C. ayyari* is only known by one study [9] conducted in India on *J. sambac*. It stated that *C. ayyari* develops at a temperature ranging from 22 and 31˚C, and a relative humidity ranging from 86.75 to 91.88%. Adults emerge at night and mate 24 hours after emergence. Females live between 9 to 12 days and mate with an average of 34 eggs.

*Corythauma ayyari* seems to thrive in the warm-summer Mediterranean climate of the European regions [38–40]. This temperate climate (minimum temperatures between -3˚C and 18˚C throughout the year) offers dry and warm summers, with precipitation less than 40 mm and maximum temperatures above 22˚C, and mild winters [39]. The period of greatest activity of *C. ayyari*, would probably occur between late spring and early autumn, coinciding with the warm summers of the Mediterranean climate, while in winter during the coldest periods, *C. ayyari* would overwinter [12, 40, 41].

The native distribution of *C. ayyari* is restricted to tropical climates (minimum temperature greater than or equal to 18˚C) of equatorial savannah with dry winter (rainfall in the driest month less than 60 mm), and to warm temperate climates with dry winter and hot summer (average temperature of the hottest month greater than 22˚C) [30, 40]. The temperature and the presence of a dry period are thus important variables common to both native and invaded areas of *C. ayyari*. Low temperatures seem to be the main constraints to the establishment of *C. ayyari*. Thermophilic in nature, it survives, like most other Tingidae, at a temperature range of 15˚C to about 40˚C, with an optimal development and fecundity rate around 30˚C [8, 10, 41–43]. At lower temperatures, it enters hibernation. The mild winter of the Mediterranean climate should therefore be favourable to it. Thus, under the worst-case scenario ssp585, temperatures will increase by 3.3˚C to 5.7˚C, for the period 2081–2100 [44]. This increase in temperature in the Mediterranean climate may facilitate optimal development of *C. ayyari*, while reducing its hibernation periods in future years.

*Jasminum* species are mainly recorded from their native area in Asia, except *J. officinale* which is mainly recorded from Europe, due to its importation as ornamental plant. Then, distribution modelling shows that Europe is mainly suitable for *J. officinale*, but weakly suitable for *J. multiflorum*, and not suitable for *J. sambac*. Mediterranean coasts only are suitable for *J. grandiflorum*.

Not surprisingly, the main suitable areas for *C. ayyari* are in the native region, South East Asia, despite few occurrences recorded there, comparing to the occurrences in Europe. Outside its original range, according to our models when considering climatic variables only, the most suitable areas for *C. ayyari* are concentrated on the North coasts of Mediterranean Sea, where it has been recently recorded, and where *J. officinale* and *J. grandiflorum* are present. Parts of North West of Europe remain also suitable for *C. ayyari*. Not surprisingly also, suitable areas are restricted to a thinner range along North Mediterranean coasts and part of Western Europe at a lower suitability, when *Jasminum* species are included as variables. This is mainly due to *J. officinale* and *J. grandiflorum* distributions, as *J. sambac* has no suitable areas in Europe, and *J. multiflorum* has a low suitability in Northern Europe. As expected, *C. ayyari* suitable area is restricted by its host plants distribution.

Projections into the future for *J. grandiflorum and J. officinale* show that suitable areas will be restricted to a narrower band along Mediterranean coasts and will also move West of Europe. In consequence, *C. ayyari* will also be restricted to the same areas, according to the projection including *Jasminum* species as variable. *Corythauma ayyari* suitable areas will extend North and East of Europe according to modelled distribution without *Jasminum* variables. That could be unconvincing, considering low temperatures in Central Europe during winter, and the near absence of *Jasminum* in this region. *Jasminum* is absent in the East of Europe, except two occurrences of *J. sambac* near Stockholm and near Kharkiv (East of Ukraine), and an occurrence of *J. officinale* near Tallinn.

Modelling of *C. ayyari* distribution and projection into the future differs when including or not *Jasminum* species as variable. *Jasminum* distribution modelling shows that *C. ayyari* will not benefit from its main host plants to spread out in Europe as would suggest modelling using only climatic variables. In addition, including the host plants as biotic variables render the modelling more accurate (see Boyce index). This show the importance of integrate biotic variables such as host plants for phytophagous species in modelling. Nevertheless, model validation is low (Boyce index around 0.7). Modelling would benefit from a larger sample size of *J. sambac* and *C. ayyari* which are around 30 occurrences each, nevertheless sufficient according to Jimenez-Valverde's criteria [28]. Small geographic and environmental ranges can also positively affect model performance [27, 45]; which seems to be the case of host plants and *C. ayyari* in the non-native range.

At the present day, the suitable area of *C. ayyari* in Europe does not fit with *J. officinale* distribution and is restricted to Mediterranean coasts, like *J. grandiflorum*. Harbours are the main entrance of the host plants, and then *C. ayyari* would have been introduced in Europe with its host plants [8]. It was recorded on *J. grandiflorum* in Spain and Syria; *J. sambac* and *J. grandiflorum* in Tunisia; *J. sambac* and *J. multiflorum* in Israel; *J. officinale* and *J. grandiflorum* in Italia. However, the invaded niche of *C. ayyari* and of *J. officinale* are similar (P value = 0.0297), while this is not the case of *J. grandiflorum*'s niche, and Europe is not suitable for *J. sambac*. As native and invaded niche of *C. ayyari* is neither equivalent nor similar, this would suggest that *C. ayyari* shifted to one niche to another, and could potentially benefit from the presence of *J. officinale* to setting up in Europe. As niche modelling shows, *J. officinale* is one of the most contributing variables (23%). More South in Mediterranea, *J. sambac* and *J. grandiflorum* could stay the main plant hosting *C. ayyari*. Unfortunately, it is difficult to know the contribution of climatic variables to niche modelling as we used PCA components to avoid auto-

correlation between variables. Another point to consider also is that the low overlap between *Jasminum* species and *C. ayyari* in the native area may be due to the lack of occurrences as no studies were conducted on the relationships between these species in Asia. Most of the records of *C. ayyari* in the native range is due to casual collecting. Overlap in Europe is mainly due to the fact that *C. ayyari* is recorded specifically as non-native species on cultivated or ornamental plants. However, Broenniman et al. [18] used kernel density function to determine a smoothed density of occurrences when estimating ecological niche. That makes moving from geographical space, where the species occur, to a multivariate environmental space, where analyses are performed, independent of sampling effort.

Niche shift of introduced plant species is not rare as stated by Atwater et al. [46]. This assumption highly depends on variable use and modelling method and has been debated on invasive plants [47, 48], and also on introduced mammals [18].

The adaptability of *C. ayyari* suggest that it could spread to other host plants. Indeed, *C. ayyari* has been observed on the genera *Lantana*, *Ocimum*, *Musa*, *Hedychium*, and *Trachelospermum*, and the species *Althea officinalis*, *Daedalacanthus nervosus*, and *Volkameria inermis* [7, 12–14]. The observation of *C. ayyari* on these plants does not mean they are host plants but may be only visited plants. Most of them are popular ornamental plants and widespread in gardens so, their trade would be the main source of the introduction of *C. ayyari* and one of the major factors for its expansion. For example, *Trachelospermum jaminoides* is probably the *Trachelospermum* species on which *C. ayyari* was found in France [11], as it is the only species of the genus recorded in Europe. It has a wide distribution in Europe and could be a potential host, facilitating *C. ayyari* spread.

## Conclusions

This study shows that host plants distribution is key to phytophagous species. They may restrict invasive potential of the invader. However, oligo- and polyphagous species have the possibility to shift to several host plants and then invade new areas, shifting from one niche to another. *Halyomorpha halys* (Stal, 1855) is a typical example of a successful invasive species as it is highly polyphagous and feeds on more than 120 wild or cultivated plant species (Streito et al. 2021). Modelling distribution depends on data availability, as invasive species provide scarce data when early considered. Therefore, modelling their potential invasiveness is often done once species already established and disrupted native ecosystem.

In several countries like India, Pakistan, Egypt, Tunisia, *Jasminum*, especially *J. grandiflorum* and *J. sambac*, have a very high symbolic value and are emblematic plants of the country (Haouas et al. 2015). They are also used in the perfume industry. The invasion of *C. ayyari* in the non-native regions would cause significant economic but also symbolic damage. Therefore, as human activity is the main source of *C. ayyari* expansion due to the ubiquitous presence of ornamental plants in gardens and parks, means of management in the latter are conceivable, and would be different from a widely cultivated plant as food resource.

## Supporting information

**S1 Data. GPS coordinates of occurrences of the species used in the analyses.** GPS coordinates of *Jasminum officinale* (Joff), *J. grandiflorum* (Jgra), *J. multiflorum* (Jmul), *J. sambac* (Jsam) and *C. ayyari* (Cay) occurrences after cleaning (see method section), used for analyses. (DOCX)

**S1 Fig.** Projections of the current (on the left) and future (on the right) environmental conditions resulting from the niche modelling for Jasminum species. Ensemble projection combines

all model with an arbitrary threshold up to 0.7 and uses the TSS binary metric, for the four main Jasminum species hosting C. ayyari: A, B) J. officinale; C, D) J. grandiflorum; E, F) J. multiflorum; G, H) J. sambac.
(TIF)

**S1 Table. Correlations between species distribution.** Correlation between species distribution using Pearson coefficient, under current condition (above) and future conditions (below). Species distributions are rasters resulting from species distribution modelling using PCA components as variables.
(DOCX)

**S2 Table. Contribution of variables used for modelling *C. ayyari* distribution.** A first modelling uses only the first nine climatic variables issued from a PCA; a second modelling includes also the four *Jasminum* species distribution as variable.
(DOCX)

## Acknowledgments

The authors thanks Alexandre Schickele for providing help with R scripts, and Jean-Claude Streito for fruitful comments on the manuscript.

## Author Contributions

**Conceptualization:** Eric Guilbert.

**Formal analysis:** Manon Durand, Eric Guilbert.

**Investigation:** Eric Guilbert.

**Methodology:** Manon Durand, Eric Guilbert.

**Supervision:** Eric Guilbert.

**Validation:** Manon Durand, Eric Guilbert.

**Visualization:** Manon Durand.

**Writing – original draft:** Manon Durand, Eric Guilbert.

**Writing – review & editing:** Eric Guilbert.

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
