## [Decision Letter · Decision Letter 0]

26 Dec 2023

PONE-D-23-37771Corythauma ayyari (Insecta, Heteroptera, Tingidae) depends on its host plant to invade EuropePLOS ONE

Dear Dr. Guilbert,

Thank you for submitting your manuscript to PLOS ONE. After careful consideration, we feel that it has merit but does not fully meet PLOS ONE’s publication criteria as it currently stands. Therefore, we invite you to submit a revised version of the manuscript that addresses the points raised during the review process.

**ACADEMIC EDITOR: **I invite you to submit a revised version of the manuscript.==============================

We look forward to receiving your revised manuscript.

Kind regards,

Faham Khamesipour, Ph.D.

Academic Editor

PLOS ONE

Journal Requirements:

3. We note that Figures 1, 2, and S1 in your submission contain [map/satellite] images which may be copyrighted. All PLOS content is published under the Creative Commons Attribution License (CC BY 4.0), which means that the manuscript, images, and Supporting Information files will be freely available online, and any third party is permitted to access, download, copy, distribute, and use these materials in any way, even commercially, with proper attribution. For these reasons, we cannot publish previously copyrighted maps or satellite images created using proprietary data, such as Google software (Google Maps, Street View, and Earth). For more information, see our copyright guidelines: http://journals.plos.org/plosone/s/licenses-and-copyright.

a. You may seek permission from the original copyright holder of Figures 1, 2, and S1 to publish the content specifically under the CC BY 4.0 license.  

Reviewers' comments:

Reviewer's Responses to Questions

**Comments to the Author**

1. Is the manuscript technically sound, and do the data support the conclusions?

Reviewer #1: Partly

Reviewer #2: Yes

2. Has the statistical analysis been performed appropriately and rigorously? 

Reviewer #1: I Don't Know

Reviewer #2: Yes

3. Have the authors made all data underlying the findings in their manuscript fully available?

Reviewer #1: Yes

Reviewer #2: Yes

4. Is the manuscript presented in an intelligible fashion and written in standard English?

Reviewer #1: Yes

Reviewer #2: Yes

5. Review Comments to the Author

Reviewer #1: Line 1-2: I don't think that this title can be chosen based on the results presented. Invasion is also not a good term to use here (it might not happen). Spread or proliferate might be better. The fact that this species depends on its host plant seems to be known so the title does not really present the results of the study.

Line 15: Invasiveness would need to be defined as a term or concept with references, ”potential to become widespread” might be better.

Line 20: What kind of modeling was done? Ensemble forecasting?

Line 20-22: Rephrase the sentence and specifically mention your species. Otherwise it reads as if it has not been considered before with other insect plant association. Or is this something really new?

Line 26: I don't think this is relevant in your study. Please give some general citations here, maybe specifically for disruptions caused by insects and/or plants.

Line 41: Maybe it would be nice to introduce the concept of pest species (in the native range) and invasive species (non-native range) once.

Line 46: Please add: introduced via trade “to Europe”. These Jasmine species are just considered a non-native species which has value as an ornamental plant and therefore should not be damaged by C. ayyari (which would be the “bad” invader)?

Line 49: Are they a real problem in the native range, thus considered a pest species? If so, why are there so little occurrence records of C. ayyari? Due to no one sampling it?

Line 53: Which other crop species could this be?

Line 61: The interesting bit is that biotic variables are rarely taken into account because the relationship to a species’ occurrence is often not entirely known. But in the case of plant and parasitic insect, the host plant’s (or many host plants’) occurrence(s) and (in)directly its climatic requirements should be the main factors that determine whether the insect survives, unless there are some other environmental factors that are different for the insects survival and the plants survival, e.g. predator of the insect. I am not sure looking into all future distribution model for the 4 Jasmine species alone is necessary for this study in such detail (see suggestion to cut part in the Results below).

Line 62: If they are strictly dependent on the host plant, why would you model only with bioclimatic variables at all? I would expect strong correlations between host plant and some climatic variables if you put them together into one model.

Line 67: Please be more specific what these studies showed.

Line 68-74: As said above, why model it without host plant at all if it is strictly dependent on it?

Line 78: I think it is debatable whether you want to call it invasion if it is considered a pest species in its native range already although there are so little number of occurrence points from its native range, which of course might be due to a sampling bias (too little sampling effort). Would the sampling (and any occurrence records) of C. ayyari not be correlated with Jasmine species?

Line 86: See comment before. Do you know whether sightings of C. ayyari were made on plants?

Line 116: Are Jasminum species already considered in equilibrium because they were introduced long time ago and are not spreading any longer in Europe?

Line 155-157: If C. ayyari cannot survive without its plant host, this analysis is not really useful.

Line 164-165: That seems very little occurrences to base models on overall. Can you comment on the minimum number of occurrence records necessary for the different models you used? If this is accounted for by using the TSS and different thresholds are calculated in each model irrespectively of what the number of occurrence records is, please describe it in more detail.

Fig.1: There is no co-occurrence in the native range? I suppose sightings were made on other host plants than Jasminum species. Do you know on what plants sightings (occurrence records) of C. ayyari were made?

Do these species not actually overlap in the native range in Asia?

Line 172: Please explain the Boyce index as a means to validate model results in more detail.

Line 170-225: This is just describing the Figures in the supplementary material and the correlation tables (which should also go into the supplementary material) focusing on the bioclimatic variables and current and future distribution for the plants and C. ayyari separately. The main focus should rather be on comparing the current and future models excluding or including these Jasmine species as biotic variables when projecting current and future distribution C. ayyari. Try to shorten it or place text into supplementary.

Line 232-236: This is the main result, incl. Fig. 2.

Line 243-244: What does “all models together” mean? The models used for the ensemble forecasting? What are these 5th, 1st and 2nd PCA components? Can you show the PCA plots?

Line 249: If this is the case, why would you expect a high invasion potential of C. ayyari due to availability of Jasmine species in Europe? Are you assuming that in the native range (Asia) they don’t overlap but in Europe they could? It seems that very little observations of C. ayyari living on Jasmine spp have been made in Asia (judging by the low amount of occurrence records on GBIF) although the insect is considered a pest species. Due to the little number of occurrences, I would also question the niche parameter calculations for C. ayyari (line 270-272). Maybe you can discuss the quality of data in the discussion.

L. 267: Could you explain the Schoener’s index meaning in more detail? When do you talk of a high overlap and a low overlap? Same for similarity and equivalency.

Line 275: What does Csa mean?

Line 278-281: It would be good to describe the life cycle of C. ayyari in more detail to see at what stage the host plant plays a role.

L 284-285: What does Aw and Cwa mean?

L. 321: Can you comment on model validation a bit more? And the data availability and how this might have influenced the projections, especially C. ayyari for which data records are very low.

L. 331-333: How reliable is it to talk of a niche switch here? How dependent is it on the host plant and how dependent on the climatic conditions?

L. 335-337: Do you want to discuss niche shift (comparing native and invaded area) of the plants as well?

L 342-344: If they are not host plants and only visited plants, could they still damage them? Have there been any experimental studies been done to test the “vulnerability” of these plants?

L 362-363: But C. ayyari is considered native in some of these regions if I understood correctly and considered a pest species. Why do you talk of its invasion in these regions then?

Reviewer #2: In the reviewed manuscript by Manon Durand and Eric Guilbert presented results of modeling the possibility of successful development of the invasive bug species of Tingidae, Corythauma ayyari in Europe depending on the distribution of its host plants, some species of jasmine.

The analysis of the possibilities of settling of this invasive species using special programs makes it possible to foresee the distribution of C. ayyari in the future. In general, the results appear to be quite reliable. However, the authors’ conclusions about the probability of significant suitability (up to 50%) for the habitat of the tropical species C. ayyari (lines in the text 205-214) in the territory of Central Europe seem unconvincing to me.

In addition, the authors' suggestion that the oligophage C. ayyari "...can damage jasmine species as well as other ornamental and/or agricultural plants, causing significant environmental and economic damage" (lines 52-44) appears to be unfounded, and this assumption follows put away

Overall, the manuscript is recommended for publication with minor corrections.

6. PLOS authors have the option to publish the peer review history of their article (what does this mean?). If published, this will include your full peer review and any attached files.

Reviewer #1: No

Reviewer #2: No

---

## [Author Response · Author response to Decision Letter 0]

31 Jan 2024

Dear reviewers and editor, We added a doc entitled "Response to reviewers" where we answered to all the comments made. You may also see in the doc with track changes what we modified according to the comments. We hope this is ok like this and that you will be satisfied by the responses.

Yours sincerely,

Eric

---

## [Decision Letter · Decision Letter 1]

14 Feb 2024

PONE-D-23-37771R1Corythauma ayyari (Insecta, Heteroptera, Tingidae) depends on its host plant to spread in EuropePLOS ONE

Dear Dr. Guilbert,

Thank you for submitting your manuscript to PLOS ONE. After careful consideration, we feel that it has merit but does not fully meet PLOS ONE’s publication criteria as it currently stands. Therefore, we invite you to submit a revised version of the manuscript that addresses the points raised during the review process.

We look forward to receiving your revised manuscript.

Kind regards,

Faham Khamesipour, Ph.D.

Academic Editor

PLOS ONE

Journal Requirements:

**Additional Editor Comments:**

I want you to respond to the comments below and revise the manuscript.

Maybe show in a PCA-plot all variables (as arrows) and their contribution, i.e. how they place along the two PCA-axes PC1 (x-axis, highest variability explained) and PC2 (y-axis, second highest variability explained).

New Table S1 in the Supplementary Material is missing.

To be corrected (Lines refer to text with Track Changes)

Line 133: since they WERE introduced

Line 140: as stated BY Jiménez-Valverde

Line 176: A niche equivalency and a niche similarity test (not tests)

Lines 311-313: It stated that C. ayyari develops at a temperature ranging from 22 and 31°C, AND a relative humidity

ranging from 86.75 to 91.88%. Adults emerge at night and mate 24 hours after emergence.

Females live between 9 to 12 days and mate WITH an average of 34 eggs.

Line 357: Jasminum in absent IN THE East of Germany (should it not say EUROPE?)

Line 368-369: Small geographic and environmental ranges CAN ALSO POSTITIVELY affect model performance

Line 423: these non-native regions

Reviewers' comments:

Reviewer's Responses to Questions

**Comments to the Author**

1. If the authors have adequately addressed your comments raised in a previous round of review and you feel that this manuscript is now acceptable for publication, you may indicate that here to bypass the “Comments to the Author” section, enter your conflict of interest statement in the “Confidential to Editor” section, and submit your "Accept" recommendation.

Reviewer #1: All comments have been addressed

Reviewer #2: All comments have been addressed

2. Is the manuscript technically sound, and do the data support the conclusions?

Reviewer #1: Yes

Reviewer #2: Yes

3. Has the statistical analysis been performed appropriately and rigorously? 

Reviewer #1: Yes

Reviewer #2: Yes

4. Have the authors made all data underlying the findings in their manuscript fully available?

Reviewer #1: Yes

Reviewer #2: Yes

5. Is the manuscript presented in an intelligible fashion and written in standard English?

Reviewer #1: Yes

Reviewer #2: Yes

6. Review Comments to the Author

Reviewer #1: All comments have been addressed.

Maybe show in a PCA-plot all variables (as arrows) and their contribution, i.e. how they place along the two PCA-axes PC1 (x-axis, highest variability explained) and PC2 (y-axis, second highest variability explained).

New Table S1 in the Supplementary Material is missing.

To be corrected (Lines refer to text with Track Changes)

Line 133: since they WERE introduced

Line 140: as stated BY Jiménez-Valverde

Line 176: A niche equivalency and a niche similarity test (not tests)

Lines 311-313: It stated that C. ayyari develops at a temperature ranging from 22 and 31°C, AND a relative humidity

ranging from 86.75 to 91.88%. Adults emerge at night and mate 24 hours after emergence.

Females live between 9 to 12 days and mate WITH an average of 34 eggs.

Line 357: Jasminum in absent IN THE East of Germany (should it not say EUROPE?)

Line 368-369: Small geographic and environmental ranges CAN ALSO POSTITIVELY affect model performance

Line 423: these non-native regions

Reviewer #2: Dear Authors,

thank you for your work in improving the manuscript, and for the answering questions. I have to notice that all statements in the article are justified, and the results of statistical analysis confirm the obtained conclusions. The presented study has theoretical and practical significance. I believe that the manuscript can be accepted for publication in PLOSE ONE journal in its current form.

7. PLOS authors have the option to publish the peer review history of their article (what does this mean?). If published, this will include your full peer review and any attached files.

Reviewer #1: No

Reviewer #2: No

---

## [Author Response · Author response to Decision Letter 1]

16 Feb 2024

Response to reviewers

Dear Editor,

Here the answers of all the comments of the reviewers. We made all changes according to reviewer requests. We also added a figure (Fig 3) showing the PCA plots of the 19 climatic variables under current and future conditions, as requested. 

Thank you for considering our revised manuscript.

Eric

Maybe show in a PCA-plot all variables (as arrows) and their contribution, i.e. how they place along the two PCA-axes PC1 (x-axis, highest variability explained) and PC2 (y-axis, second highest variability explained).

Sure! It has been done in Fig 3, in the results.

New Table S1 in the Supplementary Material is missing.

Oops… We added it. Sorry.

To be corrected (Lines refer to text with Track Changes)

All corrections hereafter have been considered in a manuscript with only these track changes

Line 133: since they WERE introduced

Line 140: as stated BY Jiménez-Valverde

Line 176: A niche equivalency and a niche similarity test (not tests)

Lines 311-313: It stated that C. ayyari develops at a temperature ranging from 22 and 31°C, AND a relative humidity ranging from 86.75 to 91.88%. Adults emerge at night and mate 24 hours after emergence. Females live between 9 to 12 days and mate WITH an average of 34 eggs.

Line 357: Jasminum in absent IN THE East of Germany (should it not say EUROPE?)

Line 368-369: Small geographic and environmental ranges CAN ALSO POSTITIVELY affect model performance

Line 423: these non-native regions

---

## [Decision Letter · Decision Letter 2]

21 Feb 2024

Corythauma ayyari (Insecta, Heteroptera, Tingidae) depends on its host plant to spread in Europe

PONE-D-23-37771R2

Dear Dr. Guilbert,

We’re pleased to inform you that your manuscript has been judged scientifically suitable for publication and will be formally accepted for publication once it meets all outstanding technical requirements.

Kind regards,

Faham Khamesipour, Ph.D.

Academic Editor

PLOS ONE

Additional Editor Comments (optional):

Reviewers' comments:

Reviewer's Responses to Questions

**Comments to the Author**

1. If the authors have adequately addressed your comments raised in a previous round of review and you feel that this manuscript is now acceptable for publication, you may indicate that here to bypass the “Comments to the Author” section, enter your conflict of interest statement in the “Confidential to Editor” section, and submit your "Accept" recommendation.

Reviewer #1: All comments have been addressed

2. Is the manuscript technically sound, and do the data support the conclusions?

Reviewer #1: Yes

3. Has the statistical analysis been performed appropriately and rigorously? 

Reviewer #1: Yes

4. Have the authors made all data underlying the findings in their manuscript fully available?

Reviewer #1: Yes

5. Is the manuscript presented in an intelligible fashion and written in standard English?

Reviewer #1: Yes

6. Review Comments to the Author

Reviewer #1: (No Response)

7. PLOS authors have the option to publish the peer review history of their article (what does this mean?). If published, this will include your full peer review and any attached files.

Reviewer #1: No

---

## [Editor Report · Acceptance letter]

16 Mar 2024

PONE-D-23-37771R2 

PLOS ONE

Dear Dr. Guilbert, 

I'm pleased to inform you that your manuscript has been deemed suitable for publication in PLOS ONE. Congratulations! Your manuscript is now being handed over to our production team.

Kind regards, 

on behalf of

Dr. Faham Khamesipour 

Academic Editor

PLOS ONE